# The Endocannabinoid System in Human Disease: Molecular Signaling, Receptor Pharmacology, and Therapeutic Innovation

**DOI:** 10.3390/ijms262211132

**Published:** 2025-11-18

**Authors:** Matei Șerban, Corneliu Toader, Răzvan-Adrian Covache-Busuioc

**Affiliations:** 1Puls Med Association, 051885 Bucharest, Romania; mateiserban@innbn.com (M.Ș.); razvancovache@innbn.com (R.-A.C.-B.); 2Department of Neurosurgery, Carol Davila University of Medicine and Pharmacy, 050474 Bucharest, Romania; 3Department of Vascular Neurosurgery, National Institute of Neurology and Neurovascular Diseases, 077160 Bucharest, Romania

**Keywords:** endocannabinoid system, CB1 receptor, CB2 receptor, gut–brain axis, immune modulation, neuroinflammation, microbiota–gut–brain connection, cannabinoid receptors, anandamide, 2-arachidonoylglycerol

## Abstract

The endocannabinoid system (ECS) is a primary regulatory system in human physiology that serves to help maintain homeostasis throughout the nervous system, immune system, and gastrointestinal system. This review has the goal of evaluating the unique opportunity for the ECS to provide a regulatory axis within the microbiota–gut–brain axis, particularly with regard to neurodevelopment, immune tolerance, and gut health. Cannabinoid receptors CB1 and CB2 and endogenous ligands anandamide (AEA) and 2-arachidonoylglycerol (2-AG have the ability to provide a variety of signaling pathways that can regulate cognitive resilience, emotional tuning, and immune regulation. Because the ECS has the ability to regulate multiple neurochemicals, alter immune cell functions, and maintain gut barriers, the ECS exists at the crossroads of many physiological systems, which also have a predictive role in neurodegenerative disease, chronic inflammation, and mental illness. Our goal is to present the latest and best recent advances in the ECS literature and establish evidence that there exists some modest potential for the therapeutic modulation of the ECS to improve pathological manifestations of cross-system dysregulation. In addition to cellular signaling pathways, the ECS affects other homeostatic processes, such as synaptic plasticity and the level of neuroprotection in the CNS, immune-related homeostasis, and coordinating the composition of gut microbiota. We argue that the ECS represents a suitable new therapeutic target that could modulate dysregulation across these systems more inclusively. This paper aims to emphasize the proposed potential of the ECS’s position in this axis and propose advanced cannabinoid-based interventions as a novel mechanism for developing personalized medicine and health systems through multi-system integration.

## 1. Introduction to the Endocannabinoid System and Its Integrative Role

### 1.1. Overview of the Endocannabinoid System (ECS)

The ECS is a fundamental regulatory system that maintains homeostasis across multiple organs, engaging functions that include neurological, immune, and metabolic endpoints. Despite the ECS’s biological relevance, the system’s complexity was not fully appreciated until the late 20th century. In the decades since, knowledge has evolved to position the ECS as an important system for neuroprotection, immune modulation, and gut–brain communication [1].

The ECS comprises three major elements: cannabinoid receptors, CB1 and CB2, endogenous ligands, anandamide and 2-arachidonoylglycerol, and metabolic enzymes, which catalyze the synthesis and degradation of endogenous cannabinoids and ECS activity [2]. CB1 receptors are expressed in the central nervous system (CNS) at areas of the brain including the hippocampus, the amygdala, and the cortex regions, where CB1 activity can alter neurotransmitter release, cognition, and emotions experienced. CB2 receptors are primarily expressed in immune cells, which help modulate inflammatory responses and maintain immune homeostasis [3,4].

There are also other receptors that involve GPR55 and TRPV channels, which may play a role in metabolic regulation and gut–brain signaling and further enhance the functional dimensions of ECS activity [5,6]. The primary endocannabinoids, anandamide (AEA) and 2-arachidonoylglycerol (2-AG), regulate ECS signaling through activation of CB1 and CB2 receptors. Anandamide, often referred to as the bliss molecule, typically displays affinity for activating CB1 receptors that regulate pain signaling, adaptation to stress, and mood [7]. Anandamide’s functional relevance is circumstantially moderated by rapid treatment (functionally, degradation) via fatty acid amide hydrolase (FAAH). Contrastingly, 2-AG interacts with both CB1 and CB2 receptors, influencing synaptic plasticity, as well as immune activity with a specific inactivator, monoacylglycerol lipase (MAGL) [8,9].

In addition to playing a critical role in the nervous and immune systems, the ECS interacts with gut microbiota constituents to contribute to several functions, including gut epithelial barrier structure, immune function, and metabolic homeostasis. Gut microbiota generate short chain fatty acids (SCFAs) that can alter the activity of the ECS; thus, gut–brain communication and signaling via the ECS can modulate responses to inflammatory processes. Dysregulation of this system is associated with disorders including neurodegenerative disease, inflammatory bowel disease (IBD), metabolic disorders, and psychiatric illness, highlighting the ECS as a therapeutic target [10].

The ECS was first described in the 1960s, when an Israeli biochemist named Raphael Mechoulam isolated the active ingredient in Cannabis sativa, Δ9-tetrahydrocannabinol (THC). Sequencing efforts led to increased understanding of the physiological functions of the endogenous cannabinoid system, and, in the late 1980s, scientists identified CB1 and CB2 receptor subtypes as molecules that modulate the activity of the ECS. Subsequently, endogenous cannabinoids were discovered in the early 1990s. Since its identification, the ECS has been shown to be involved in many critical physiological functions, including pain perception, neuroprotection, metabolism, and immune regulation [11,12].

The ECS is central to maintaining a range of homeostatic physiological conditions because it integrates signals within complex interdependent systems (neurological, immune, metabolic).

To assess the role of the ECS as a functional bridge across neuroscience, immunology, and microbiota research, a table (Table 1) summarizing 25 important studies published in the last decade was created. The table summarizes the main aspects of research that integrated the ECS’s influence within its interdependent physiological systems across a range of applications related to protective nervous functions (i.e., neuroprotection) or immunological functions (i.e., immune modulation), gut–brain communication, and therapeutic potential. The studies are presented to illustrate the progression of ECS research and the emergence of novel mechanistic and therapeutic insights. The table summarizes key aspects of each publication, including the model system, ECS targets, investigated mechanisms, and the therapeutic relevance of the findings. This summary will provide useful information describing the path the research trajectory has followed towards addressing an important and new area of clinical research, demonstrating the potential for cannabinoid-based therapies that may contribute to new approaches for treating complex multi-system diseases.

The ECS has recently been documented to play roles in metabolism, hormonal mediation, and circadian alignment. It is an energy balance reactor system that regulates appetite, glucose homeostasis, and deposition of lipids. It is the conductor of hormonal orchestras that regulates the hypothalamic–pituitary–adrenal (HPA) axis and reproductive health [37]. The ECS also interacts with circadian regulatory mechanisms, aligning biological processes with day–night cycles and maintaining temporal homeostasis across neural, immune, and metabolic systems. The ECS has demonstrated utility, even in the context of SARS-CoV-2 impacting the world, providing insight into how immunity can be modulated and even demonstrating antiviral behavior [38].

### 1.2. The Triad of Neuroscience, Microbiota, and Immunology

The ECS is a regulator of cross-talk between the nervous, immune, and gastrointestinal systems. This implies that the ECS is capable of integrating interactions among systems to regulate stable physiological homeostasis. The ECS represents a connection between the neurological, immune, and gastrointestinal systems as a functional and regulatory unity [39].

Neurospecific functions include CB1 receptors, which play roles in synaptic plasticity, the regulatory influence of neurotransmitter production, and neurogenesis. The ECS helps maintain cognitive functioning and the plasticity of neurons by modulating excitatory/inhibitory signaling to limit excessive excitation, which can relate to impairments like epilepsy [40]. Both ECS and the microbiota–gut–brain axis, which includes the HPA stress pathway, regulate stress and immune responses. While the HPA axis operates through systemic hormonal feedback loops, the ECS modulates stress and immune reactivity via localized, lipid-based signaling within neural, immune, and gut circuits [41,42,43].

The ECS is also a maintainer of immune homeostasis through interactions with immune cells’ CB2 receptors, especially in macrophages, T cells, and microglia that modulate the intensity of immune and inflammatory responses. ECS-mediated signaling can also persist in limiting excessive inflammatory/immune responses that support tissue repair and immune tolerance [44]. CB2 receptor effects in the CNS can help manage neuroinflammation and excessive immune responses that can exacerbate neuroinflammatory processes [45,46].

Given its modulatory roles in the neuroregulatory, immune, and metabolic domains, the ECS will become an improved therapeutic target in complex disorders like Alzheimer’s disease, IBD, and mood disorders. Dysregulation of ECS signaling has been reported in disease processes across organ systems, suggesting a critical space for intervention [47].

### 1.3. Objectives of This Review

The ECS will be reported on as a key regulator of neuroscience, immunology, and microbiota interactions, which together integrate dynamics of signaling within these systems. This review provides an integrated analysis of its structural and functional character, paying particular attention to the cellular mechanisms that mediate cross-system interactions. In addition to reporting novel connections in signaling of the ECS, including the lesser-known receptors, systems approaches will be articulated focusing on emerging research opportunities in ECS’s involvement in developing systems-mobilized immune responses, again relating that argument back to viral immune responses like those associated with SARS-CoV-2. Then, we will clearly identify conditions that could be improved through ECS interventions directly related to disorders where neuroinflammatory complaints, immune dysregulation, and metabolic imbalance are themes.

Integration of a multi-disciplinary perspective will provide opportunities for growth of knowledge but also further information beyond the boundaries of this review. The goal is to move towards an improved understanding of the ECS’s potential for intervention in disease and develop viable innovative therapeutic pathways.

## 2. Cannabinoid Receptors: Convergence Point of the Triad

Cannabinoid receptors are the molecular fulcrum of the ECS, facilitating cross-talk between neuroimmune and gastrointestinal systems. The main receptors, cannabinoid receptor 1 (CB1R) and cannabinoid receptor 2 (CB2R), are classical binding sites for endogenous and exogenous cannabinoids. Other cannabinoid-related receptors, such as GPR55, GPR18, and TRP channels, modulate the functional dimension of the ECS, which, although differing in signaling downstream in each subtype and tissue, allows for homeostatic physiological functioning across systems. These receptors are abundant in the CNS, but their expression in peripheral and enteric tissues would suggest homeostatic functional integration between neurophysiology, immunity, and the microbiota [48].

### 2.1. Cannabinoid Receptor Type 1 (CB1R)

CB1R, a G-protein coupled receptor (GPCR), is one of the most abundant GPCRs in the mammalian brain and localized primarily to areas controlling higher-order functioning, such as the hippocampus (memory and learning), the basal ganglia (motor control), the cerebellum (co-ordination), and the prefrontal cortex (decision making) [49,50]. Activation of the CB1R is also dependent on depolarization, which is generated directly by activity and involves vintage signaling in which endocannabinoids (2-arachidonoylglycerol [2-AG] and anandamide [AEA]), generated postsynaptically, diffuse back to the presynaptic terminal to inhibit neurotransmitter release, particularly glutamate and GABA, which control excitatory and inhibitory synaptic transmission in these areas, thus modulating synaptic plasticity, including long-term potentiation (LTP) and long-term depression (LDP) [51,52].

CB1R is also highly expressed in the enteric nervous system (ENS) outside of the CNS, where it modulates visceral nociception, motility, and permeability of the epithelial lining. Dysregulated CB1R signaling in the ENS has also been implicated in the pathophysiology of functional gastrointestinal disorders, such as irritable bowel syndrome (IBS), where dysfunctional gut–brain communication would be involved in the underlying mechanism of visceral pain and dysmotility [53].

Within the context of the immune system, the expression of CB1R is found in microglial and astroglial cells, although lower than CB2R, but it allows for anti-inflammatory modulation because it is capable of suppressing interleukin-1β (IL-1β) and tumor necrosis factor-α (TNF-α). In addition, CB1R is involved in the modulation of the hypothalamic–pituitary–adrenal (HPA) axis in order to attenuate stress-induced hyperactivation by maintaining systemic homeostasis [54].

### 2.2. Cannabinoid Receptor Type 2 (CB2R)

CB2R shows low expression in the CNS from baseline but is abundant in tissues of immune and peripheral natures (macrophages, T cells, B cells, and dendric cells), where it regulates inflammatory balance [55]. Its activation results in a decrease in the concentration of pro-inflammatory cytokines (IL-6, TNF-α) and increases anti-inflammatory mediators, such as interleukin (IL)-10, which generate the stability of immune homeostasis and a decrease in autoimmunity [56].

CB2R is highly inducible during pathological inflammatory processes. In neurodegenerative diseases like Alzheimer’s disease, upregulation of the receptors in microglial and astroglial cells is able to reduce oxidative stress and neuroinflammation, which results in neuronal survival [57]. This has rendered CB2R an important therapeutic target for autoimmune and neuroinflammatory disorders [45,58].

In the gastrointestine, CB2R may act in conjunction with immunological functions and the integrity of the epithelium of the intestine through potentiation of the proteins of tight junctions (occludin, claudin-1), which bind to enhance barrier function to prevent the translocation of antigens or microbes. This protective function is important in IBD, which includes Crohn’s disease and ulcerative colitis. The signaling from CB2R may also effect a change in the composition of the physiological microbiota of the gastrointestinal tract and is able to mediate structurally and physiologically bidirectionally the interlocking interactions so that both immune tolerance and systemic inflammatory processes can be affected. Cannabinoid receptor type 2 (CB2R) is emerging as a therapeutic target for the treatment of infections like SARS-CoV-2, where activation of the receptor may lessen the release of cytokines from and promote reparative processes in damaged tissues (and thus potentially lessen morbidity and mortality) [59,60].

### 2.3. Putative Cannabinoid-Related Receptors

The cannabinoid receptors GPR55 and GPR18 and the TRP channels add to the complexity of ECS signaling. GPR55, formerly classified as an orphan receptor and showing a partial overlap in activity with CB1R/CB2R, binds to a number of endocannabinoids. Among them, lysophosphatidylinositol has been shown to mediate pro-inflammatory signaling in macrophages among other immune cells and recruitment of leukocytes and is involved in osteoarthritis, induction of cancer metastasis, and other inflammatory conditions [61,62].

The expression of GPR18 is found on several types of immune and endothelial cells; a key endogenous ligand for this receptor is N-arachidonylglycine (NAGly). The activation of GPR18 is involved in regulating the movement of immune cells into tissues, blood vessel constriction/dilation, and the level of inflammation. GPR18 agonists have been shown to prevent endothelial dysfunction and inhibit leukocyte infiltration in atherosclerosis, which demonstrates its potential use as a therapeutic target in reducing the inflammatory component of cardiovascular disease [63,64].

In addition to the above-mentioned effects of GPR18, it is involved in the modulation of mood through its participation in the gut–brain axis. Changes in the sensitivity of the GPR18 receptor may be caused by changes in levels of microbiota-derived metabolites, such as indoles, which are produced through alterations in the balance of the microbial composition of the gastrointestinal tract; these alterations may contribute to the development of stress-related mood disorders. GPR18 agonists administered to mice were shown to decrease stress-related behaviors, stimulate the generation of neurons in the hippocampus,, and normalize the microbial composition of the GI tract, which further supports its role in modulating the immune system and brain function [65,66].

The TRP channels, particularly the TRPV1 channel, historically identified with nociceptive and warmth sensing, can also be modified by the cannabinoid AEA, among others. TRPV1 abnormal functioning has been associated with pain responses to visceral stimulation and motility dysfunction, as well as IBD, which may be present in symptomatic subjects. TRPV1 functional dysregulation represents a possible target class of therapies [67,68].

Allosteric modulation of the CB1R and CB2R receptors appears to be a fertile area for therapeutic targeting. Allosteric modulators alter receptor activity through non-orthosteric modulation and titration of activity, avoiding side effects associated with respective full agonists or antagonists. This approach may have applications in chronic pain, metabolic syndrome, and diseases of neurodegeneration [69].

### 2.4. Receptor Localization in Microbiota and Gut Barrier Function

The ECS is an important part of maintaining gut barrier integrity and host interaction with the microbiota. The receptors CB1R/CB2R are associated with expression within the intestinal epithelial cells as well as regulation of junction proteins, such as zonula occludens-1 (ZO-1), which regulate intestinal epithelial permeability, as well as translocation of bacterial endotoxins (lipopolysaccharide, LPS, for instance), which reflect a protective strategy, in order to avoid systemic inflammatory response to obesity and metabolic syndrome and septic-related consequences [70,71,72].

The microbial metabolites, principally the short-chain fatty acids, also reflect a means through which there is modulation of the copy number and the synthesis of receptor component proteins and signaling, which engenders a feedback system and thus may integrate microbial messaging with respect to ECS-influenced regulatory-related pathways. Disruption of this microbiota–ECS interaction would appear responsible for disorders classed as psychiatric (depression, anxiety), metabolic (obesity), or inflammatory (IBD-related) [73].

Overall, it appears that cannabinoid receptors possess and act on gateways to complex intracellular signaling systems. These are at work with respect to physiological intranetworking network adaptation across systems. Sequentially, the next section will deal with endocannabinoid signaling systems and signaling pathways, demonstrating the respective interconnections that are thus able to act downstream from exogenous input stimuli to cellular functional processes and appropriate responses, illustrating the complexity of the system and correlating therapeutic benefits the ECS may provide [74].

## 3. Endocannabinoid Signaling in the Neuroscience–Microbiota–Immunology Axis

The ECS serves as a mediator, integrating the CNS, the immune system, and the gut microbiota. It modulates interactions between these systems through biochemical signaling that maintains homeostasis despite internal or environmental insults. Through its effect on the microbiota and immune processes, the ECS modulates both synaptic plasticity in the brain as well as barrier function in the gastrointestinal tract. While the ECS has been mostly recognized as a retrograde mechanism of neurotransmission, it also functions as a paracrine and lipid signaling system mediating communication between the brain and peripheral organs. All in all, it is a regulatory system that is fluid with respect to neural, immune, and gastrointestinal responses in order to maintain physiological balance.

### 3.1. Mechanisms of ECS Signaling

#### 3.1.1. Retrograde Signaling in the CNS and the Gut

ECS retrograde signaling is a signature complication of ECS involving the process of moving from classical neurotransmission. Classical neurotransmission consists of forward signaling by presynaptic neurons releasing neurotransmitters to postsynaptic neurons as the only method of communication. Retrograde signaling is the process whereby postsynaptic neurons produce endocannabinoids (2-AG and anandamide) as required based on fluctuations in calcium levels within postsynaptic neurons. Looking at a postsynaptic neurons with feedback versus classical neurotransmission, postsynaptic neurons control the presynaptic terminal with feedback to some extent. In the end, feedback from a postsynaptic terminal decreases presynaptic action potential induced through neurotransmitter release by first inhibiting the action of presynaptic voltage-gated calcium channels (VGCCs) and facilitating the presynaptic inwardly rectifying potassium channels for fast synaptic change [75]. Retrograde signaling does not discount postsynaptic signaling nor a presynaptic cell’s involvement, which is very important for future PFC to perform different forms of retrograde signaling.

The retrograde signaling aspect of the ECS has critical consequences for synaptic plasticity and results in substantial effects on LTP and LTD, which are needed for memory consolidation, learning, and modulating behavior. For example, in the specific case of endocannabinoid and E/I balance in the hippocampus, endocannabinoids can suppress excess glutamate release while preferentially allowing more GABA signaling. This fluid regulation is important for effective synaptic function and to prevent excitotoxicity [76,77].

This retrograde signaling pathway is also exhibited in the gastrointestinal (GI) system, and the impact of retrograde signaling can be observed in the neuroimmune interactions of the ENS related to gut motility, secretion, and visceral sensitivity. Endocannabinoids modulate the release of neurotransmitters from enteric neurons regulating peristaltic contractions and secretory responses, which are key to an effective digestive process. The adverse effects of malfunctioning retrograde signaling, characterized by the dysregulation of gut motility and ‘visceral hypersensitivity’ associated with IBS, provide credible evidence that the ECS-based gut–brain communication pathway has physiological and clinical relevance [78].

#### 3.1.2. Autocrine and Paracrine Signaling in Immune Cells

Another example of ECS-mediated regulation of immune responses that occur in a more local manner is autocrine and paracrine signaling. In the presence of inflammation, immune cells, including macrophages, dendritic cells, and T cells, synthesize endocannabinoids that act at the local level to reduce immune responses [79]. Endocannabinoids can elicit local responses to inflammation through two types of signaling: autocrine or paracrine. In autocrine signaling, the endocannabinoid(s) produced will act locally on the immune cell that produced it, and the endocannabinoids will bind to CB2 and other receptor subtypes, which can repress the pro-inflammatory transcription factors (e.g., nuclear factor-kappa B (NF-κB)) and activate the anti-inflammatory pathway (e.g., peroxisome proliferator-activated receptor gamma (PPAR-γ)). In paracrine signaling, endocannabinoids are released from their cell of origin, diffusing to nearby cells and having an effect on their activity, which serves to synchronize the resolution of inflammation [80].

Paracrine-mediated signaling in the gastrointestinal tract is key to enterocyte and gut-associated lymphoid tissue (GALT) homeostasis. The activation of CB1R and CB2R increases the expression of the tight junction proteins ZO-1 and claudin-1 in epithelial tissues, resulting in an increase in the epithelial barrier properties against translocation of microbes across the gut wall and systemic inflammation [81]. Besides increased structural integrity, ECS-mediated signaling is also involved in neuroimmune and paracrine-mediated inflammation of the gastrointestinal tract. More specifically, the activation of CB1R decreases the release of inflammatory neuropeptides (substance P and calcitonin gene-related peptide (CGRP)), leading to decreased neurogenic inflammation and visceral hypersensitivity. CB2R is expressed on immune cells, such as macrophages, dendritic cells, and intraepithelial lymphocytes, and decreases the release of TNF-α, IL-1β, and IL-6 whilst positively regulating IL-10 secretion. This leads to an anti-inflammatory and tolerogenic immune microenvironment. ECS-linked enzymes also regulate local levels of inflammation; inhibition of FAAH leads to increases in anandamide levels and decreased neutrophilic infiltration and oxidative stress, while blockade of MAGL leads to elevated levels of 2-AG, a potent inhibitor of leukocyte activation and release of inflammatory cytokines. Therefore, the associated mechanisms show that the ECS-induced modulation of intestinal inflammation is not limited to a maintaining function of tight junctions but includes many aspects of neuroimmune signaling, homeostasis of the pro-/anti-inflammatory cytokine milieu, and paracrine metabolism and regulation, resulting in multifactorial action in the repair of intestinal barrier dysfunction [82].

#### 3.1.3. ECS and the HPA Axis

The ECS and HPA axis relationship can be drawn out of ECS’s function in modulating stress response employed throughout the body. For example, activation of CB1R in the hypothalamus inhibits the release of corticotropin-releasing hormone (CRH), which then inhibits the release of adrenocorticotropic hormone (ACTH) from the anterior pituitary and then blocks the release of cortisol from the adrenal glands. This negative feedback displays the EC and HPA response to acute stress; however, with chronic stress, the downregulation of endocannabinoids, and the ability to activate their signaling using the CB1R, one could theoretically be hyperactive in their HPA axis, leading to prolonged cortisol production, which can propagate neuroinflammation and cause maladaptive synaptic plasticity [83,84]. Hence, stress dysregulation could be implicated in mood disorders, such as anxiety, depression, and post-traumatic stress disorder (PTSD), and the ECS could be a therapeutically transferable target for reinstating an individual’s stressed HPA axis to homeostatic function [85].

Figure 1 illustrates the complexity of the ECS signaling cascade, depicting the molecular components and their interrelations within neuronal synapses.

### 3.2. Signal Transduction Pathways in ECS Signaling

#### 3.2.1. G-Protein-Coupled Receptor (GPCR) Signaling

CB1R and CB2R are both ordinary G-protein-coupled receptors (GPCR) and initiate a range of intracellular events through Gi/o protein signaling. Endocannabinoids activate the GPCR, causing activation of Gi/o protein; through activation of the Gi/o protein, the α-subunit of the Gi/o protein inhibits adenylyl cyclase, which lowers cyclic AMP (cAMP) levels and inhibits protein kinase A (PKA) activity. This suppression of CB2R modulation influences intracellular calcium dynamics, thereby decreasing neurotransmitter release at the terminals of neurons and cytokines produced by immune cells [86,87].

In neurons, CB1R activation produces MAPK cascades (involving ERK1/2, JNK, and p38 MAPK), altering synaptic remodeling and neuroprotection as well as adaptive cellular responses to stress. In immune cells, MAPK signaling via CB2R diminishes inflammatory mediators and aids tissue repair, illustrating the role of the ECS in immune resolution [88]. The ECS is able to directly control and modulate ion channels and receptor activity, providing a localized and rapid, albeit transient, change in excitability at a cellular level. For example, CB1R activation decreases VGCCs presynaptically to regulate calcium influx to regulate the release of neurotransmitters. Furthermore, CB1R opens potassium channels, hyperpolarizing the neuron and dampening potential action excitability, mostly by stabilizing the synaptic network and protecting neurons from excitotoxic damage.

The ECS also regulates peripheral ion channel function; this includes the action of AEA on transient receptor potential vanilloid 1 (TRPV1), which has a role in the modulation of pain perception as well as heat regulation by virtue of its presence in various physiological states. TRPV1-mediated signaling within the GI tract is involved in the mediation of visceral sensation, motility, and mucosa protective functions; thus, ECS–TRPV1 dysfunction is implicated in both chronic pain syndromes as well as irritable bowel syndrome [89,90].

#### 3.2.2. ECS Integration in Lipid Signaling

The previous discussion of the ECS indicates that its integration into important lipid signaling pathways causes the ECS to behave more like a metabolic regulator [91]. Anandamide is produced from N-arachidonoyl-phosphatidylethanolamine (NAPE) via NAPE-specific phospholipase D (NAPE-PLD) and ultimately broken down by FAAH. 2-AG follows the same pathway from diacylglycerol (DAG), also forming from DAGL via hydrolysis by MAGL [92]. The pathways indicated are unique to the ECS and produce the major endocannabinoids, but they also intersect with different bioactive lipid pathways involved in inflammation, vascular tone regulation, and the production of acute pain [93,94].

FAAH catalyzes the breakdown of anandamide in cells, and when FAAH activity is elevated, it is associated with a decrease in overall anandamide levels and shown to result in stress sensitivity and emotional dysregulation. In contrast, inhibition of MAGL and increased 2-AG signaling can produce beneficial neuroprotective and anti-inflammatory activity in models of neurodegeneration and traumatic brain injury.

### 3.3. ECS Dysregulation and Disease States

ECS dysfunction appears to drive chronic disease states within the body; the lack of signaling drives synaptic downregulation, immune dysregulation, and breakdown of epithelial barriers. For example, neurodegenerative diseases, such as Alzheimer’s or Parkinson’s disease, demonstrate the links between ECS dysregulation and accelerating synaptic degeneration, neuroinflammation, and, possibly, oxidative stress [95]. Likewise, in the gastrointestinal tract, dysfunctional ECS signaling in IBD was shown to break down normal epithelial barrier function, thus sustaining gut inflammation. The ECS habitat is strongly implicated in metabolic diseases and insulin resistance, obesity, and systemic inflammation [96].

Therapeutic modulation of the ECS occurs through its integrated control over three major body systems, i.e., the nervous, immune, and gastrointestinal systems. The main ways to modulate the ECS include using CB2 receptor agonists to suppress neuroinflammation; using FAAH and MAGL inhibitors to elevate levels of endocannabinoids; and microbiota-directed treatments to enhance gut–brain–immune equilibrium [97].

In addition, emerging genetics and biotechnology strategies will also provide new options for the interventions described above. For example, CRISPR/Cas9-based genetic modification of the FAAH gene may be used to repair loss-of-function mutations of this gene and thus restore anandamide signaling in individuals with neurodegenerative disease. Similarly, genetically modified probiotic bacteria that produce endocannabinoid analogues are being investigated as a potential therapy to restore balance to the ECS in the gastrointestinal tract and treat systemic inflammation and disorders associated with dysbiosis [98].

The ECS operates via varied signaling mechanisms and thus operates as a homeostatic cornerstone and not just as part of disease pathology. In the next section, we will focus on how these signaling pathways contribute to both physiological and pathological states, including how they shape neurodevelopment, synaptic plasticity, and interactions between the gut, brain, and immune systems.

## 4. Neuroscience: The ECS at the Intersection of the Microbiota and Immunology

At the center of influence in the world of neuroscience is the ECS, a modifier of neurocircuitry, a modifier of immunity, and a regulator of neurocommunication of the gut, as well. While it was previously known to modulate neurotransmission and synaptic activity, recent evidence has identified that the ECS is involved in neurodevelopmental, neuroprotective, and gut communications [97]. This section reviews its integrative roles in cell development and systemic regulation to illustrate its therapeutic and biological importance.

### 4.1. The ECS in Neurodevelopment and Synaptic Plasticity

The ECS is functional in embryogenesis, where the presence of the CB1R in the neural progenitor cell (NPCs) enhances proliferation, migration, and differentiation. CB1R signaling represses adenylate cyclase and Gi/o protein predominating in pathways that modulate Wnt/β-catenin signaling and CREB signaling, which determines NPCs’ fate. Also under the influence of CB1R is cytoskeletal remodeling, which through RhoA-ROCK signaling enhances neuronal migration and axonal elongation, which are important for creating the cortical and subcortical circuitry [99]. From the signal epigenetically is CB1R, which alters chromatin architecture through histone acetylation and methylation at loci like Sox2 and NeuroD1. The maladaptive changes occurring in this vital developmental period enhance the risk of development of disorders like autism spectrum disorder (ASD) and schizophrenia, and, in this context, the ECS represents an option for molecular scaffolding of neuroarchitecture [100,101].

With respect to what has been written about endocannabinoid signaling, the important issue is synaptic plasticity, which is the cellular basis of learning and memory. Endocannabinoids, such as 2-AG and anandamide, are produced in a localized area to allow for retrograde action to inhibit presynaptic release of glutamate and GABA and maintain the normal balance of excitation–inhibition [102]. The role of CB1R within the hippocampus is critical to long-term potentiation (LTP) and LTD—two processes essential for memory consolidation. ECS signaling through astrocytes modulates the expression of EAAT2, which is necessary for limiting glutamate-induced excitotoxicity. The CB1R also aids in supporting myelination in oligodendrocytes, which is important to support action potential conduction along axon segments in an effective and stable manner [103]. Overall, these findings indicate that the ECS is a modulator and stabilizer of neural network activity [104,105].

### 4.2. The ECS in Neuroinflammation and Neurodegeneration

The immune cells of the CNS, microglia, are key effector cells of neuroinflammation and highly susceptible to modulation by ECS signaling. Under conditions of homeostasis, microglia constantly survey neural tissues for inflammation and tissue injury [106]. Upon activation, CB2R shifts microglia to an anti-inflammatory phenotype, upregulating IL-10 while downregulating IL-6 and TNF-α [107]. CB2R also rewires microglial metabolism from glycolysis to oxidative phosphorylation, resulting in reduced toxic reactive oxygen species and enhanced clearance of amyloid-β and cellular debris [108]. Another receptor in the ECS, GPR18, promotes directed microglial migration and phagocytosis in the presence of neuronal injury, further contributing to CNS repair [63].

Dysregulation of the ECS contributes to neurodegenerative disorders, such as Alzheimer’s and Parkinson’s disease. CB1R signaling ameliorates glutamate-induced excitotoxicity and induces the expression of various anti-apoptotic proteins (e.g., Bcl-2), while CB2R induces enhanced autophagic clearance of aggregates of tau and α-synuclein [109]. CB1R also promotes mitochondrial stability through interaction with the mitochondrial calcium uniporter, resulting in a stable mitochondrial membrane potential and prevention of mitochondrial depolarization. CB2R enhances blood–brain barrier (BBB) integrity, leading to a decrease in immune cell infiltration to the CNS. Collectively, these findings describe the ECS as a target and regulator of neurodegeneration [96].

### 4.3. Psychiatric Disorders and the Gut–Brain Axis

The ECS regulates mood, emotional resilience, and adaptation to stress, which are intimately linked to the gut–brain axis. The anandamide level is directly correlated with the activity of the HPA axis; activation of CB1R inhibits the release of corticotropin-releasing hormone and the release of downstream cortisol, decreasing chronic stress levels, but the synaptic levels of anandamide are low with decreased expression of CB1R and correlate with psychopathology, including anxiety, depression, and post-traumatic stress disorder (PTSD) [110]. Microbial metabolites stemming from the gut (e.g., short-chain fatty acids (SCFAs), butyrate) affect the ECS by leading to increased expression of CB1R and CB2R in the frontal cortex and hippocampus and provide a mechanistic rationale for microbiota-directed treatment of mood [111]. Inhibition of FAAH, which degrades anandamide, also increases the tone of endocannabinoids and thus normalizes gut microbiota towards a balance associated with stress, suggesting bidirectional therapeutic effects [112].

ECS effects also display circadian variations. Daily variations in the expression of CB1R in the suprachiasmatic nucleus synchronize the activities of neurons to light–dark cycles. Deviations in daily patterns (as with shift work or jet lag) contribute to dysregulation of the HPA axis and instability in mood. Therapies that modulate the ECS in belief of its pattern might enhance affective resilience [113].

### 4.4. Emerging Therapeutic Frontiers

Recent investigations have shed light on the mechanistic role of the ECS in the epigenetic regulation of neural plasticity. Stimulation of CB1R evokes histone acetylation in the promoters of the neurotrophic genes, such as BDNF, that enhance memory consolidation and functionality after injury [114]. The phenomenon of receptor heteromerization provides terrific diversity to ECS signaling. Thus, CB1R demonstrably heterodimerizes with TRPV1 in sensory components, leading to effects in pain and inflammation. GPR55–CB2R complexes are noted to be present in microglial elements, permitting dual points of therapeutic intervention in neuroprotection and neurorepair [99,112,115].

These modalities suggest that the ECS has an integrative capacity, referring to neurological, psychiatric, and immunological situations, and allows for the development of dual-agonist or microbiota effects companies in the larger area of personalized medicine. From a systems perspective, the ECS is responsible for processes of neurodevelopment, synaptogenesis, anti-inflammatory processes, and gut–brain communication such that it serves as a molecular substrate for reparative and adaptive processes. Its next roles can be seen in gut–brain–immune intercommunications and translational applications of its benefits to the health of the body.

## 5. Microbiota: The ECS as a Mediator of Gut–Brain–Immune Communication

The ECS provides for communication between microbiota, the immune system, and the nervous system, enabling the conversion of microbial signal responses into host physiological responses. This bidirectional form of communications in terms of homeostasis and protection against inflammatory and metabolic alterations is functional.

### 5.1. Influence of Gut Microbiota on ECS Activity

The level of microbial metabolites is greatly influential on the ECS in terms of receptors and enzymes (and, in some cases, receptors) for all ligands. Also, fatty acid products, derivates of tryptophan, and secondary bile acids regulate CB1R and CB2R, forming and degrading the endocannabinoids, such that homeostatic intestinal repair is maintained [116,117].

The products of indole (tryptophan metabolism) produced by bacteria are responsible for the stimulation of anandamide production and stimulants to vagal afferents downstream such that there is a connection between microbial metabolism and regulation of emotions and stress. In germ-free species, it is observed that there is a decrease in cortical levels of anandamide and respiration related to increased stress reactivity and an inability to regulate emotional aspects of life [118].

The effects of some of the metabolites derived from the microbiota also communicate through CB2R in gut-associated lymphoidal tissue (GALT), where there is modulation of macrophagic and dentic cell functions towards modifications that alter the responsiveness of the immune system towards tolerance. Increased dysbiosis is seen in Crohn’s disease, which gives rise to the imbalance that occurs, which alters mucosal inflammatory responses [119].

### 5.2. ECS Regulation of Gut Barrier Function

The ECS controls the balance of gut barrier integrity by means of its regulatory effects on tight junction proteins, such as the zonula occludens-1, and also claudins, where it prevents the translocation of antigens from the gut [120]. While the stimulation of epithelial healing and regeneration is related to CB1R, CB2R is closely linked with the inhibition of pro-inflammatory cytokines and stimulation of the synthesis of IL-10, which can be viewed as a means whereby recovery of the tissue is mediated [121,122,123,124].

The ECS is linked to the stimulation of mucous production from goblet cells, permitting colonization of microflora and protection of epithelial cells. Short-chain fatty acids (SCFAs) derived from the diet enhance this function, while high doses of saturated fats diminish the density of CB receptors and alter the integrity of the barrier [125,126,127,128].

These interactions indicate the importance for systemic and gut health of feeding, while attaching dietary–microflora–ECS together. This is seen in Figure 2, indicating the interaction of microflora, mediators of immune function, and the systems of the nervous system, including, in particular, the vagus nerve, with regard to the production of the ECS as well as that of the HPA axis, indicating the link between diet and resistance to stress.

### 5.3. ECS and Probiotic/Prebiotic Interventions

Interactions of probiotics, prebiotics, and the ECS maintain gut barrier integrity and modulate inflammatory states. Particular strains of Lactobacillus and Bifidobacterium induce increases in levels of the endocannabinoid anandamide in intestinal tissues, resulting in enhanced epithelial resistance and reductions in visceral hypersensitivity via CB1R-dependent pathways [129]. Metabolites derived from probiotics also induce signaling via the CB2R on macrophages, leading to decreases in the release of pro-inflammatory cytokines, such as TNF-α and IL-6, but increases in the production of IL-10, which supports immune homeostasis. Prebiotic fiber sources, such as inulin and fructooligosaccharides, hydrate, ferment, and provide substrates for short-chain fatty acids, particularly butyrate, which acts a histone deacetylase inhibitor, inducing both CB1R and CB2R expression in conjunction with tight-junction expressed genes, maintaining epithelial integrity. Synbiotic supplementation of probiotics with prebiotics enhances both microbial diversity in the gut and CNS tone, decreasing colonic permeability and inflammatory activity [130]. Pharmacological augmentation of ECS tone can add further efficacy to these approaches. Inhibition of the enzymes FAAH and MAGL leads to elevations in levels of endogenous cannabinoids, restoration of tight junction functionality, and decreased levels of neutrophil infiltration in experimental models of acute colitis. Combination strategies targeted to the ECS and the gut microbiota would therefore appear to produce coordinated barrier-protective and anti-inflammatory effects, representing a possible experimental paradigm for future translational studies.

### 5.4. Novel Mechanisms in Microbiota–ECS Interactions

The microflora and the ECS can thus work together to present a unitary homeostatic apparatus of interaction, which can be modulated by circadian frequencies. It would in addition now appear that the microbial metabolites are capable of altering the diurnal release of CB1R in gut tissues, thus permitting ECS activity to be synchronized with appetitive periodicities. Loss of circadian synchronization caused by erratic sleep or diet induces changes to gut barriers and the release of systemic inflammatory mediators, thus showing the all-important, changing nature of chronobiology, indicating the necessity of taking this into consideration when devising remedial strategies in ECS dysregulation [131,132,133].

Still further, the microflora can influence the epigenetics of ECS elements, with butyrate-enhanced histone acetylation levels at the CB1R and CB2R gene promoter sequences being responsible for an increase in levels of expression of these receptors in epithelial and immune cells. This shows the way in which microbial diversity can dynamically change the level of expression of the host organism’s genes and alter its responsivity of ECS [109].

In conclusion, the ECS mediates communication between the ECS, the gut, and the brain–immune axis, modulating the microbial and immunological signals into physiological homeostasis. Through appropriate maintenance of the purity of the barrier, regulation of the nature of the microbial ecology, and alteration of the levels of inflammation, the ECS presents us with a useful framework for understanding complex disease states and hence the development of rational medicines for the promotion of systemic health.

## 6. Immunology: ECS-Modulated Immune Responses in the CNS and the Gut

The ECS is the conductor of the immune system, managing cellular activity across inflammation, resolution, and tissue repair in order to maintain overall health. By integrating localized immune responses with systemic signaling networks, the ECS orchestrates molecular regulation of cellular interactions, ensuring that inflammatory activity remains controlled while preserving the body’s capacity for protective immune activation. In the very immunologically active environments of the CNS and the gut, the ECS guides immune activity to maintain homeostasis and manage inflammation and resolves immune dysfunction. This section addresses the ECS’s role in immune modulation, its effects on autoimmune diseases, and its emerging role in tumor immunology.

### 6.1. The ECS in Immune Cell Regulation

The ECS has many layers of influence on immune cells, balancing anti-inflammatory and pro-inflammatory immune system activity. CB2 receptors, mostly expressed on immune cells, are the master regulators of this balance, and CB2 activation prevents inflammatory processes through inhibition of inflammatory pathways, such as nuclear factor-kappa B (NF-κB), MAPK, and Janus kinase-signal transducer and activator of transcription (JAK-STAT), indirectly inducing cytokine storms while promoting immune tolerance and tissue repair by resolving the pro-inflammatory response [88,134,135].

Some of the most influential immune cells, macrophages, are affected by CB2 signaling and by promoting the transition from the pro-inflammatory phenotype (M1) to the reparative phenotype (M2), which is mediated by metabolic reprogramming. During this metabolic reprogramming, CB2 activation enhances fatty acid oxidation and mitochondrial biogenesis while reducing glycolysis, which promotes inflammation [136]. Viewing these changes from a regulatory perspective, a great benefit of this metabolic switch is that it promotes regeneration in the tissues and protects a macrophage’s ability to survive long anti-inflammatory responses, which is necessary for chronic diseases that exhibit inflammation [137].

In T cells, the function of CB2 signaling helps to modulate the balance of pro-inflammatory T subsets, such as Th1 and Th17, with regulatory T cells (Tregs) through regulation of two transcription factors, T-bet and RORγt, suppressing the pro-inflammatory portion of Th1 and Th17. Additionally, if regulation of FoxP3 expression occurs, this will enhance Treg expansion. This is also a possible explanation for the ECS’s involvement in inflammatory diseases, such as multiple sclerosis and systemic lupus erythematosus, that have the potential to push out Th1/Th17 acceleration [138,139,140].

More recently, we have started to isolate non-coding RNA, especially microRNAs (miRNA), that can mediate ECS function as modulators of the immune system. The ECS also has the ability to induce miR146a when activated by CB2, and it has a highly significant suppressor ability on the NF-κB signaling pathway by suppressing the inflammatory cytokine production ability of TNF-α and IL-6 [141]. The most notable of these miRNA mediators is miR-155, which inhibits T cell dysregulation and macrophage activation with some pro-inflammatory potential. The miRNA portion of this process is able to account for another dimension of the ECS’s immune-modulatory influence and suggests that there is possibly an epigenetic influence associated with immune cells [142,143].

With regard to the CNS, the resident immune cells in microglia are in an energetic environment and respond to ECS signaling. Activation of CB2 may alter pro-inflammatory cytokine production from microglia and enhance the secretion of neurotrophic factors (BDNF) that could possibly translate to enhanced synaptic repair and provide protection against the inflammatory damage of the surrounding neuronal network [144]. Particular developmental influences of the ECS include modulating the microglial phagocytosis of cellular debris and toxic protein aggregates and, ultimately, downregulating inflammatory cascades through the degradation of cellular pattern recognition receptors, which stimulate the neurodevelopmental cascades that produce neurodegenerative effects, speeding up their influence on acceleration [57].

Table 2 intends to be an all-encompassing summary of original studies describing the ECS. We reviewed significant findings across various domains, including anxiety, pain insensitivity, drug discovery, and established ECS interactions with the gut microbiota. The table is divided into five columns: study reference, research focus, methodology, findings, and outcomes. Each study entry briefly describes how the research contributes to the current understanding of the role of the ECS in health and disease and, ultimately, the therapeutic possibilities.

Figure 3 provides a visual representation of the way cannabinoids bind to these receptors, illustrating that the ECS has a dual functionality in neurons and immune cells.

### 6.2. The ECS in Autoimmune and Inflammatory Diseases

The ECS is essential in regulating autoimmune and inflammatory diseases by regulating interactions between different immune cell subsets and tissue-specific responses. In autoimmune diseases, such as multiple sclerosis (MS), the ECS reduces CNS inflammation by inhibiting the migration of autoreactive T cells across the blood–brain barrier. CB2 activation attenuates chemokine signaling, limits T cell invasion on patrolling inflammatory cells, and decreases astrocyte and microglia release of inflammatory mediators, such as IL-1β, TNF-α, etc. In preclinical models of MS, CB2 agonists display a decrease in demyelination and neuronal loss, demonstrating that the ECS has therapeutic possibilities in neuroinflammation [162,163,164].

In the gastrointestinal tract, the ECS maintains intestinal homeostasis by regulating the immune responses that underlie IBD. CB1 and CB2 signaling promote epithelial barrier function through tight junction proteins (occludin and ZO-1) that limit microbial translocation and prevention of systemic inflammation that leads to colitis. Evidence has shown that CB2 activation can reduce inflammatory cytokines released from macrophages and T cells, which would lead to tissue repair and reduced severity in IBD. Clinical trials testing cannabinoid-based therapies in patients with Crohn’s disease and ulcerative colitis have shown efficacy when testing treatments to impact outcomes of disease and markers of disease activity, which support therapeutic targeting of the ECS in gut inflammation [53,165,166,167].

The role of ECS modulation is important in systemic autoimmune diseases like systemic lupus erythematosus (SLE). One example of this is the use of CB2 agonists, which reduced hyperactivation of B cells and inhibited pathogenic autoantibodies that promote the tissue damage that drives SLE. Furthermore, the ECS could promote Tregs’ ability to inhibit autoreactive T cells and restore immune tolerance to reduce the number of flares in the disease.

### 6.3. Role in Tumor Immunology

The ECS is capable of facilitating tumor-associated immunity in the highly dynamic and hostile tumor microenvironment. By activating CB2 on tumor-associated macrophages (TAMs), the phenotype can flip from a tumor-supportive M2-like state to an M1-like anti-tumor state while simultaneously promoting recruitment and activation of cytotoxic T cells and inhibiting the release of pro-tumor cytokines like IL-10 and transforming growth factor-beta (TGF-β). The ECS can effectively remodel the tumor microenvironment and destroy immunosuppressive barriers/modulations/reservations that tumors depend on for immune evasion [168,169,170].

Cannabinoid-based therapies can also target/impose immune checkpoint pathways tumors use to inhibit T cell activity. Recent studies have suggested that ECS signaling may alter PD-L1 expression in tumor cells, thus altering PD-L1 activity as a method to inhibit T cell activation. CB1 signaling can alter angiogenesis via endothelial cells, thus removing the vascular supply tumors utilize for growth and metastases [171,172,173].

ECS-targeted strategies have also been looked at from an agricultural perspective, as it has been reported that some cannabinoids can affect plant stress responses and pest resistance behaviors. A series of recent studies have demonstrated that cannabinoid analogs can be applied to crops to improve drought response capabilities via lipid signaling pathways similar to those underlying ECS functioning in humans. As an aside, from an agricultural point of view, further studies specific to ECS biology are also forthcoming, and these results present exciting opportunities to harness ECS-based research for sustainable agriculture [174,175].

The implications of the ECS for tumor immunology continue to expand to include potential synergy with already existing treatments. By modulating the ECS early and reducing inflammation, it may be theoretically possible to enhance immune cell infiltration into tumors, particularly while employing immune checkpoint inhibitors and adoptive T cell therapies. The ECS is also an attractive target for integrative cancer immunotherapy due to its potentially multi-faceted effects, as well as its already established effectiveness in mitigating common side effects in cancer patients [176].

### 6.4. Future Technologies: Advancing ECS Research

The ECS, as an active and developing area of interest in biomedical research, is also at the verge of innovative studies with tools like artificial intelligence (AI), machine learning, and multi-omics approaches. The emergence of AI/machine-learning-based approaches has the potential to revolutionize how we study ECS functioning in health and disease and enables a deeper, more collaborative approach when studying ECS functions. There is exciting potential for these new technologies to transform ECS-focused therapeutic discovery, as they can utilize precision medicine, where medications, including ECS-targeted drugs, can be matched to patients’ unique phenomena based on individual patients’ distinct molecular make-up as it pertains to ECS function. New and exciting applications utilizing AI and machine learning in ECS fields have begun.

#### 6.4.1. Artificial Intelligence in ECS Drug Discovery

Advances in artificial intelligence (AI), machine learning (ML), and the resulting predictive algorithms have accelerated the process of finding drugs that target the ECS. The binding affinity of potential candidate ligands for CB1 and CB2 receptors can be modeled using predictive algorithms, thus enabling researchers to select compounds with desirable pharmacodynamics at an earlier stage in the drug discovery process. The use of AI-based virtual screening and molecular docking technology significantly reduces the need for expensive laboratory experiment; this technology identifies selective CB2 agonist drugs with anti-inflammatory capabilities and significantly reduced risk of psychoactivity [177,178].

Furthermore, the dynamic behavior of ECS receptors in the context of immune cell function has been studied using AI-driven modeling approaches, and these studies have informed the development of therapeutic treatments for diseases including multiple sclerosis, lupus, and inflammatory bowel disease [179].

#### 6.4.2. Multi-Omics and AI in ECS Research

Multi-omics methods (transcriptomics, lipidomics, and metabolomics) are providing a more comprehensive understanding of ECS signaling mechanisms and identifying new biomarkers and therapeutic targets. Transcriptomics studies have shown that CB2-dependent gene expression occurs in the context of cytokine signaling, apoptosis, and immunomodulation [180,181,182].

Lipidomics studies have demonstrated the interaction between endocannabinoids and SCFAs, which are important for the resolution of inflammation and tissue repair [183,184]. Additionally, metabolomics studies have identified that certain microbiota-derived metabolites (short-chain fatty acids, bile acids) modulate ECS activity and link the type of microbiota present to immune tolerance and gut homeostasis [185,186,187].

The integration of AI with multi-omics will enable researchers to predict how patients will respond to treatment on an individual basis. Deep learning models are currently being developed that can take genomic, epigenetic, and metabolic data together to define ECS signaling patterns in the neural, immune, and gut systems. These models will ultimately lead to precision ECS-based therapies and biomarker-guided clinical trials [188,189].

#### 6.4.3. Biotechnology and Engineered Solutions

In addition to biotechnologically engineered probiotics capable of producing endocannabinoid analogs to restore ECS tone in the gut and counteract the detrimental effects of dysbiosis-induced inflammation, other engineered therapeutic platforms also exist. For example, nanoparticles used to deliver cannabinoids and ECS active agents to tissues provide improved specificity to specific areas of the body, minimize systemic side effects, and increase therapeutic efficacy [190,191].

Collectively, the convergence of biotechnology, engineering, AI, and multi-omics is transforming ECS research and its translational potential. This convergence provides a platform for developing personalized ECS interventions that consider the interplay among the neural, immune, and microbiota systems in a unified therapeutic approach.

## 7. Genetics and Epigenetics: Shaping ECS Function in the Triad

The ECS is shaped by genetic variation and epigenetic plasticity in neural, immune, and digestive systems. Genetic polymorphisms create the architecture for receptors, enzymes, and ligands, while epigenetic effects create an ability to respond dynamically to environmental signals, producing susceptibility to disease and treatment effects.

Polymorphisms involving CNR1 and CNR2 directly modulate receptor efficiency. The CNR1 polymorphism rs1049353 has been associated with decreased responsiveness of the CB1 receptor (CB1R), leading to decreased control of emotional and somatic stresses. This may increase susceptibility to IBS [192]. The polymorphism CNR2 rs35761398 decreases signaling through the CB2 receptor (CB2R) of the immune system and thereby decreases anti-inflammatory resilience, leading to an increased susceptibility to autoimmune diseases, including rheumatoid arthritis and multiple sclerosis [193]. There are also polymorphisms in enzymes that modulate endocannabinoid tone. FAAH rs324420 decreases responsiveness of the enzyme and increases levels of anandamide, which improves resilience to stress while increasing susceptibility to drug abuse and metabolic dysregulation. There are also polymorphisms that modulate the availability of 2-AG and thus neuroprotective and anti-inflammatory properties [194,195,196]. Epigenetic modification is a second tier in the modulation of ECS function. The methylation of DNA in genes’ codes for the ECS is genetically programmed to regulate transcription in conformation with environmental pressures. Chronic stress gives rise to hypermethylation of the CNR1 promoter, leading to decreased levels of CB1R expressed in brain areas involved in emotion and stress resilience, effects contributing to anxiety and depression [197]. On the other hand, hypomethylation of CNR1 in certain cancers leads to increased expression of CB1R, which may aid in tumorigenesis, pointing to the possibility that methylation modulates ECS tone across a number of diseases [198].

The interplay between histone modifications further adds to this adaptability. Activation of CB1R gives rise to increased levels of histone acetylation in key neuroprotective genes, such as BDNF, while activation of CB2R serves to recruit histone deacetylases (HDACs) to inhibit pro-inflammatory transcription during immune recovery [199,200]. At the post-transcriptional level, microRNAs modulate ECS expression; miR-146a upregulation through CB2 inhibits the NF-κB pathway via TRAF6 and IRAK1, while miR-155 inhibition reduces autoimmune inflammation [143,201]. Environmental influences continuously affect this genomic and epigenomic environment. Nutrition, stress, and the gut microbiota act together to reorganize ECS functioning. Omega-3 fatty acids provide substrates for endocannabinoid synthesis and increase ECS loci histone acetylation, increasing CB1/CB2 expression in the brain and the gut. Polyphenols, like green tea and berry elements, act as HDAC inhibitors to enhance the ECS’s anti-inflammatory actions [202,203].

Chronic stress downregulates the ECS through CNR1 hypermethylation and a decreased supply of anandamide, which are reversible through FAAH inhibition [194]. The gut microbiota provides another layer of epigenetic control; metabolites, such as butyrate and propionate, both HDAC inhibitors, increase CB1/CB2 expression in gut tissues, while dysbiosis curtails all regulation of this sort and leads to systemic inflammation. This can be counteracted by microbial rebalancing through probiotic and prebiotic treatments, which therefore represent an indirect action of such treatments to normalize ECS function.

These interwoven genetic and epigenetic mechanisms form the basis of the science of pharmacogenomics, which aims to optimize ECS-targeted medicine. CNR1, CNR2, and FAAH polymorphisms can serve as predictive markers of response; for example, patients with reduced FAAH activity may respond to CB blockades, while carriers of CNR2 pro-inflammatory polymorphisms may respond to CB2 selective agonists [204,205]. Among the other epigenetic therapeutics to be investigated, including HDAC inhibitors and miRNA mimetics, may also be found the means for targeted correction of ECS derangements in chronic inflammation, neurodeterioration, and autoimmunization [206]. Combinations of several multi-omics systems, such as transcriptomics, lipidomics, and metabolomics, combined with AI-enhanced predictive models will provide ECS networks in extraordinary detail with the ability to design individualized therapies [207,208]. These data illustrate that the ECS is a biologically flexible, environmentally adaptive system that has a capacity for molecular modification, setting the stage for individualized therapies.

## 8. Translational Applications of ECS-Targeted Therapies

With the ECS’s ability to coordinate neural, immune, and gastrointestinal signaling, the ECS gives a unifying platform for translational medicine. ECS modulation restores homeodynamics to inflammatory, neurodegenerative, and psychiatric states that are not amenable to single-pathway modulation. ECS activation in neurodegeneration has neuroprotective and reparative effects. In Alzheimer’s disease (AD), for instance, activation of CB2 on the microglia has the effect of upregulating the amyloid-β clearance systems and redirecting the microglia to the phagocytic, anti-inflammatory phenotype, while CB1 leads to a reduction in glutamate excitotoxicity and sustenance of synaptic plasticity [209]. Modulation of receptor CB1 affects not just glutamate reception but also glucose uptake and mitochondrial homeostasis. Clinical trials of THC and CBD indicate both neuroprotection and symptomatic improvement [210]. In Parkinson’s disease (PD), CB2 induction leads to reduced glial inflammatory reaction in the substantia nigra, whereas CB1 protects dopaminergic neurons from excitotoxicity and oxidative stress [211]. Allosteric modulators of the ECS are now allowing for tuning of the ECS with targeted precision measurements and improvements in side effects, which may also modify α-synuclein aggregation, an important pathophysiological event in PD [212,213]. In amyotrophic lateral sclerosis (ALS), it has been shown that CB2 reduces glial cytokinic release, while CB1 has an important role in preserving synaptic transmission, which is most useful in symptomatic upkeep. Improvements in liposomal and intranasal delivery systems are facilitating CNS localization and consequently improving patient outcomes [214,215].

In autoimmune disease, ECS modulation also restores homeodynamics to immune responses. In multiple sclerosis (MS), CB2 reduces autoreactive T cell infiltration and CB1 facilitates remyelination, providing benefits in addition to conventional immunotherapies [216,217]. In IBD, CB1 acts to maintain the integrity of the epithelial barrier, and CB2 acts to inhibit macrophage cytokine production; new strains of engineered probiotic organisms, which generate endocannabinoid analogues, in conjunction with short-chain fatty acid (SCFA) supplementation, greatly increase effectiveness [68,218,219]. In systemic lupus erythematosus (SLE), CB2 agonists downregulate autoreactive B cells and inflammatory cytokines, while CB1 signaling pathways regulate immune metabolism and the induction of Treg cells. Clinical trials are presently evaluating ECS modulators in refractory SLE cases [220,221]. In the area of mental health, ECS restoration improves emotional resilience. ECS impairment in the prefrontal cortex and the amygdala impairs stress coping and reward signaling. Inhibitors of FAAH cause increases in anandamide and have shown anxiolytic and antidepressant effects, while CBD has shown anxiolytic effects with few adverse effects [115,222,223]. In post-traumatic stress disorder (PTSD), CB1 activation promotes fear extinction and synaptic plasticity, which improves coping with trauma. CBD may improve exposure therapies, and ECS modulation, according to neuroimaging techniques, is being used as a method for providing individualized therapy [224,225]. Among the new frontiers in neuro-therapeutics are the allosteric modulators, which affect CB1/CB2 outcomes without producing psychoactive effects, and bioengineered probiotic organisms capable of synthesizing endocannabinoid analogues, which facilitate dual correction of dysbiosis and ECS inadequacy and are particularly relevant to gut–brain axis disorders [226,227,228]. Overall, ECS therapies have shown the potential of the ECS to serve as a cross-domain integrator of inflammation regulation, neuroprotection, and immune recalibration

## 9. The ECS as a Central Integrator in Health and Disease

The ECS functions as a molecular integrator of systems that interconnect nerves, immunity, and the gastrointestinal tract into a cohesive physiological spider web [177]. It enables cross-talk among these systems to maintain homeostasis and adaptive flexibility.

Within the gut–brain–immune axis, the CB1 receptor regulates psychoneurotransmission, mood, and cognition while regulating motor activity, secretion, and barrier permeability in the intestine [117,229]. Likewise, decrements in CB1 signaling can account for manifestations of IBS and functional disorders in general [230].

The microbiota alters ECS tone by virtue of metabolites that upregulate CB1/CB2, such as SCFAs, which enhance the gut barrier and attenuate systemic inflammation. Furthermore, the dysbiotic state promotes chronic inflammatory states that account for anxious and depressive states and neurodegeneration, both of which can be enhanced by probiotics and ECS-directed therapies [231,232]. The ECS also acts as the synchronizer of immune and neural products; CB2 activation on microglia inhibits neuroinflammation, whereas ECS signaling in gut-associated lymphoid tissue (GALT) maintains immune tolerance [233,234]. In situations of ECS disequilibrium, one finds multi-system disorders. Neurodegenerative, psychiatric, autoimmune, and metabolic diseases are characteristic of ECS dysregulation. In Alzheimer’s disease, Parkinson’s disease, and multiple sclerosis, one detects chronic neuroinflammation and altered synaptic stability characteristic of an abnormal ECS governing factor control. PTSD and depression are characterized by lowered endocannabinoid levels and impaired emotional readjustment [47,235]. Autoimmune diseases in which IBD and SLE manifest have aberrations in CB2 signaling associated with the maintenance of inflammation, while the crass over-activation of CB1 in the periphery leads to the development of obesity, insulin resistance, and Type II Diabetes Mellitus [236,237]. The correction of ECS tone would thus be considered stakeholders in a convergence of multi-system disorders.

Future directions for precision ECS medicine will entail the fields of pharmacogenomics and epigenetics and the use of artificial intelligence in the field of pharmacological advances. Genetic information on CNR1/CNR2 variants and epigenetic information on regulation of the ECS will provide individualized means of reestablishing balance and homeostasis in patients [238]. Combination modalities linking modulation of ECS controls with other modalities of immunological therapies, microbiota-directed therapies, and metabolic control measures will all account for the directed therapy that achieves the most clinically [129]. New technological advances with nanoparticle delivery and computer-directed ligand design will become useful in directing improved specificity, safety, and transfer of knowledge from physiological decipherment to the patient [239]. The ECS will thus stand not merely as one functional signaling characteristic but rather as an architecture providing balance to the various systems of the body, along with the capacity for linkages of molecular precision and the holistic design of therapeutic modalities [96].

## 10. Conclusions: The ECS as a Central Regulator of Systemic Homeostasis

The ECS is a global regulatory mechanism that enables communication between nervous, immune, and gastrointestinal systems. The nature of the signaling mechanism permits refinements of neurotransmission, immune functions, and integrity of digestive barrier systems to ensure stability under different environments. When the ECS is dysfunctional, it contributes to neurodegenerative diseases, inflammatory diseases, metabolic diseases, and psychiatric diseases, thus supporting the contention that the ECS is the master regulator of the resiliency of the system and also responsible for the vulnerability of the system to the development of disease. It has recently become clear that the ECS is not a discrete biochemical pathway but rather a multi-functional platform that integrates information from environmental, genetic, and micro-organismic sources into specific intracellular respondents.

The various receptors, ligands, and metabolic enzymes associated with the ECS act in unison to promote homeostatic balance within the dynamic systems that comprise the biological environment. Understanding the genetic and epigenetic regulation of the components of the ECS provides the basis for a new paradigm in personal medical solutions that aligns the molecular system components with specific therapeutic approaches for individuals. As a result of this advance in translational science, modulation of the endocannabinoid system is now providing a basis for innovative integration within biomedicine. Selective modulators of CB1 and the CB2 receptors, allosteric modulators, and inhibitors of enzymes, such as FAAH and MAGL, may emerge as modalities for the restoration of physiological balance without interference with systematic consistencies. It is the eventual potential merger of the use of ECS drugs with immunomodulator, neuroprotective, and microbiota-specific directed therapeutic modalities that represents a major milestone in the discussion of integrated therapy based on these mechanisms.

In summary, the ECS presents the opportunity to appreciate how modern biology is reconstituting the definition of health—not as an absence of disease but in promoting maintenance of the homeostatic ability of the organism to interact with heterogeneous systems. It is the future exploration of these interactions that will expose the molecular logic of the interregnum of the brain, the body, and the immune system and support modern therapeutics’ increased efficacy and a more integrated response.

## Figures and Tables

**Figure 1 ijms-26-11132-f001:**
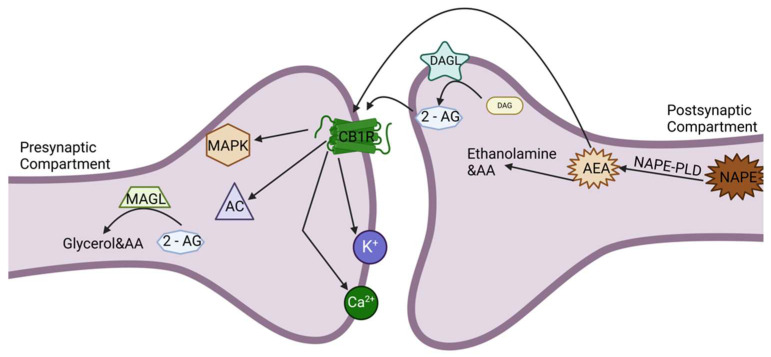
Schematic representation of the endocannabinoid signaling pathway in neurons, highlighting the principal components and their interactions. The two main endocannabinoids, 2-arachidonyl glycerol (2-AG) and anandamide (AEA), are synthesized and released in response to cellular stimuli. 2-AG is produced from diacylglycerol (DAG) through the action of diacylglycerol lipase (DAGL), while AEA is synthesized from N-arachidonoyl phosphatidylethanolamine (NAPE) via N-arachidonoyl phosphatidylethanolamine-phospholipase D (NAPE-PLD). These endocannabinoids activate cannabinoid receptor 1 (CB1), leading to downstream signaling cascades that involve adenyl cyclase (AC) inhibition and MAP kinase (MAPK) pathway activation. The enzymatic breakdown of AEA and 2-AG is mediated by fatty acid amide hydrolase (FAAH) and monoacylglycerol lipase (MAGL), respectively, resulting in the release of arachidonic acid (AA). This complex network of enzymes and receptors regulates key neuronal processes and modulates synaptic transmission.

**Figure 2 ijms-26-11132-f002:**
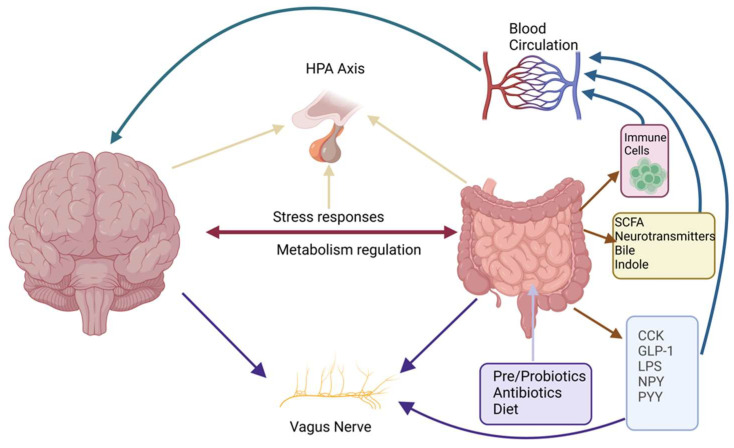
Representation of the microbiota–gut–brain axis; schematic illustration of bidirectional communication between the microbiota, brain, and gut and their role in regulating metabolic homeostasis and stress response in the context of ECS signaling. It illustrates how interactions between neural (vagal nerve), immune, and endocrine systems are connected with ECS components—cannabinoid receptors (CB1, CB2) found in the enteric nervous system, immune cells, and central stress circuitry—to modulate neuroinflammation, gut permeability, and hypothalamic–pituitary–adrenal (HPA) axis function. Metabolites produced by the microbiota (short-chain fatty acids, SCFAs; bile acids) can also modulate the tone of the endocannabinoid system, while diet, prebiotics, postbiotics, and antibiotics have been shown to shape the reciprocal communication networks between the microbiota and endocannabinoid-system-mediated feedback loops. Cholecystokinin (CCK); glucagon-like peptide-1 (GLP-1); lipopolysaccharide (LPS); neuropeptide Y (NPY); polypeptide YY (PYY); short-chain fatty acids (SCFA).

**Figure 3 ijms-26-11132-f003:**
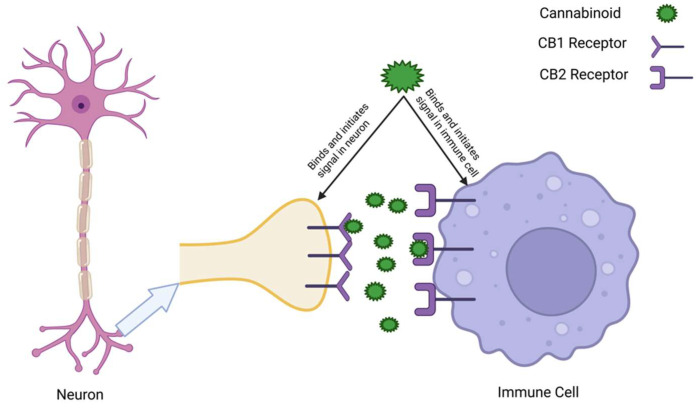
The diagram illustrates how cannabinoids can activate both CB1 and CB2 receptors, facilitating communication between the nervous and immune systems. This interaction underscores the ECS’s role in maintaining physiological balance by modulating neural activity and immune responses through distinct yet interconnected pathways.

**Table 1 ijms-26-11132-t001:** This table aims to present a curated collection of 25 significant studies on the ECS, highlighting its diverse roles across neuroscience, immunology, and gut microbiota research. Each entry includes the focus area, model system, key findings, ECS components involved, explored mechanisms, and relevance to therapeutic applications. Collectively, these studies illustrate the ECS’s integral function in coordinating cross-system communication, modulating immune responses, supporting neuroprotection, and influencing gut health. Moreover, it underscores the ECS’s potential as a therapeutic target, particularly in treating complex conditions like neurodegenerative diseases, autoimmune disorders, and mental health issues, where multi-system regulation is essential. Abbreviations: ECS—endocannabinoid system; CB1—cannabinoid receptor type 1; CB2—cannabinoid receptor type 2; TRPV1—transient receptor potential vanilloid type 1; SCFA—short-chain fatty acid; HPA—hypothalamic–pituitary–adrenal; LTP—long-term potentiation; LTD—long-term depression; GI—gastrointestinal; IBD—inflammatory bowel disease; MS—multiple sclerosis.

References	Focus Area	Key Findings and Implications	ECS Components	Mechanisms Explored	Relevance to Therapeutics
Neuroscience and Neuroprotection					
[13]	Neurodegeneration	ECS modulation reduces neuroinflammation and supports cognitive function	CB1, CB2	Microglial phagocytosis, neuroprotection	Neuroprotective strategies in Alzheimer’s disease
[14]	Neuroprotection	ECS reduces neurodegeneration and slows MS progression	CB2	Immune modulation, neuroprotection	Neuroprotective approaches in multiple sclerosis
[15]	Neurodegenerative Disease	CB2 enhances amyloid-beta clearance and limits inflammation	CB2	Microglial activation control	Alzheimer’s disease therapy
[16]	Brain Development	CB1 influences neural progenitor differentiation and cortical patterning	CB1	Neurodevelopmental signaling, Wnt/β-catenin pathway	Developmental neurobiology insights
[17]	Synaptic Plasticity	ECS modulates synaptic strength and supports LTP/LTD balance	CB1	Synaptic transmission modulation	Cognitive enhancement and memory therapies
[18]	Cognitive Health	ECS improves cognitive flexibility and reduces stress-induced impairment	CB1	HPA axis regulation, synaptic modulation	Stress resilience and cognitive therapy
[19]	Mood Regulation	CB1 regulates emotional response and stress resilience	CB1	HPA axis modulation, amygdala activity	ECS-based anxiolytic and antidepressant therapies
Microbiota–Gut–Brain Axis					
[20]	Microbiota–Gut–Brain	SCFAs modulate ECS signaling, enhancing gut–brain communication	CB1, CB2	SCFA–ECS cross-talk	Gut–brain therapies for mental health
[21]	Gut–Brain Axis	ECS regulates gut motility and visceral inflammation	CB1	Enteric signaling modulation	Gut–brain axis-targeted therapies
[22]	Gut–Immune Interaction	ECS mediates immune tolerance and reduces dysbiosis-driven inflammation	CB2	Microbiota–immune signaling	Microbiota-based immune therapies
[23]	Gut Motility	CB1 activation regulates peristalsis and gut microbial composition	CB1	Enteric nervous system signaling	GI motility and IBS interventions
[24]	Gut–Immune Balance	SCFAs induce CB2 upregulation, supporting immune tolerance	CB2	SCFA–CB2 interaction	Gut-microbiota-linked immune regulation
Immunology and Autoimmune Regulation					
[25]	Immunology	Cannabinoids modulate T cell polarization and cytokine balance	CB2	Cytokine modulation, immunosuppression	Autoimmune and inflammatory disease therapies
[26]	Autoimmune Disorders	CB2 reduces immune cell infiltration and promotes myelin protection	CB2	T cell migration control	Multiple sclerosis treatment pathways
[27]	Autoimmunity	CB2 activation suppresses T cell activity and inflammatory cytokines	CB2	Immune regulation	Autoimmune disease intervention
[28]	Autoimmunity	CB2 downregulates inflammatory cytokines and restores immune tolerance	CB2	Cytokine modulation	Therapeutic immune regulation
[29]	Immune Regulation	CB2 stabilizes cytokine networks and maintains immune homeostasis	CB2	Cytokine balance mechanisms	Anti-inflammatory therapy design
Gastrointestinal and Metabolic Health					
[30]	Gut Barrier Function	CB1/CB2 maintain epithelial integrity and reduce permeability	CB1, CB2	Tight junction stabilization, anti-inflammatory signaling	Barrier reinforcement in IBD
[31]	Inflammatory Bowel Disease	ECS signaling suppresses inflammatory cytokines and restores mucosal health	CB1, CB2	TNF-α/IL-1β inhibition	Novel IBD therapies
[32]	Metabolic Health	CB1 modulates adipogenesis and metabolic regulation	CB1	Lipid metabolism control	Metabolic syndrome treatment
[33]	Obesity and Metabolism	CB1 influences lipid storage and insulin sensitivity	CB1	Lipogenesis and glucose regulation	Anti-obesity pharmacological targets
Pain, Stress, and Epigenetic Modulation					
[34]	Pain Management	CB1–TRPV1 co-activation reduces chronic pain perception	CB1, TRPV1	Pain desensitization, neurotransmission control	Chronic pain relief
[35]	Chronic Pain	TRPV1–CB1 interaction mitigates neuropathic pain	CB1, TRPV1	Receptor cross-talk modulation	Neuropathic pain treatment
[36]	Epigenetic Modulation	Stress-induced CB1 methylation alters ECS adaptability	CB1	DNA methylation, histone modification	Epigenetic strategies in stress disorders

**Table 2 ijms-26-11132-t002:** An overview of experimental studies that elucidate the molecular mechanisms of ECS-mediated immune regulation, further supporting the therapeutic potential of CB1 and CB2 modulation. ECS—endocannabinoid system; CB1/CB2—cannabinoid receptor types 1 and 2; FAAH—fatty acid amide hydrolase; FAAH-OUT—FAAH-associated noncoding RNA gene; 2-AG—2-arachidonoylglycerol; EMR—electromagnetic radiation; CRH—corticotropin-releasing hormone; BDNF—brain-derived neurotrophic factor; DAG—diacylglycerol.

References	Focus	Methods	Key Findings	Implications
[145]	ECS in anxiety-like behavior due to EMR	Mouse model, dual-frequency EMR exposure, CB1 agonist/antagonist treatment, corticosterone/CRH levels.	Dual-frequency EMR reduced CB1 expression, lowered 2-AG levels, and induced anxiety; CB1 agonist alleviated anxiety.	CB1 agonists may counteract EMR-induced anxiety disorders.
[146]	FAAH-OUT and pain insensitivity	Genetic analysis, endocannabinoid level measurements, BDNF expression in fibroblasts.	FAAH-OUT deletion reduced FAAH, increased anandamide, and elevated BDNF expression.	FAAH-OUT deletion underpins pain insensitivity; potential for new pain treatments.
[147]	Endocannabinoid reuptake inhibitors	Chemical probe synthesis, in vivo behavioral assays, ECS activity measurement.	Novel inhibitors increased ECS signaling and produced cannabimimetic effects in mice.	Promising ECS modulators for treating ECS-related disorders.
[148]	ECS drug discovery	Screening ECS-targeting compounds for potency, receptor selectivity, and therapeutic effects.	Identified selective ECS modulators with minimal off-target effects.	New ECS drugs for inflammation, pain, and metabolic conditions.
[149]	Guineensine as an ECS uptake inhibitor	Pharmacological profiling, uptake inhibition assays, behavioral tests in mice.	Guineensine effectively inhibited endocannabinoid uptake, enhancing ECS signaling and producing cannabimimetic behavioral effects in mice.	Potential new lead for ECS-targeted therapy.
[150]	ECS in epilepsy	Mouse epilepsy models, CB1/CB2 pharmacological modulation, ECS component profiling.	ECS modulates seizure thresholds; CB1 activation reduced seizures, while CB2 regulated inflammation.	ECS-targeted therapies may improve epilepsy management.
[151]	Caffeine–CB1 receptor interaction	Animal models, CB1 receptor assays, striatum neurotransmitter analysis.	Caffeine increased CB1 activity and altered striatal neurotransmitter levels.	Insights into caffeine’s impact on ECS-mediated neural signaling.
[152]	CB1 and social behavior	CB1 expression in starling brains, behavioral observation, correlation analysis.	CB1 expression correlated with social dominance and behavior.	ECS involvement in social hierarchies and behavior regulation.
[153]	Antimicrobial properties of ECS compounds	Screening cannabis-derived compounds for bacterial inhibition, including antibiotic-resistant strains.	Cannabis compounds showed potent antimicrobial effects, especially on resistant strains.	Basis for developing ECS-derived antibiotics.
[154]	ECS and gut microbiota interactions	Fecal microbiota transplantation in mice, ECS modulation analysis, microbiome composition measurement.	ECS activity modulated gut microbiota, influencing stress responses and intestinal inflammation.	ECS–microbiota interplay as a therapeutic target for gut and mental health disorders.
[155]	ECS modulation in neuroinflammation	In vitro studies on microglial cells, CB2 receptor agonist treatment, inflammatory cytokine measurement.	CB2 activation reduced pro-inflammatory cytokine release in microglial cells.	Targeting CB2 receptors may offer therapeutic strategies for neuroinflammatory conditions.
[156]	ECS involvement in metabolic syndrome	Clinical study measuring endocannabinoid levels in patients with metabolic syndrome, correlation analysis.	Elevated endocannabinoid levels correlated with increased insulin resistance and adiposity.	ECS modulation could be a potential therapeutic approach for managing metabolic syndrome.
[157]	Role of ECS in osteoarthritis pain	Animal model of osteoarthritis, intra-articular injection of CB1/CB2 agonists, pain behavior assessment.	Activation of CB2 receptors reduced pain behaviors in osteoarthritic animals.	CB2 receptor agonists may serve as novel analgesics for osteoarthritis pain management.
[158]	ECS and stress-induced immunosuppression	Human study assessing endocannabinoid levels before and after acute stress, immune cell activity measurement.	Acute stress increased endocannabinoid levels, which correlated with reduced immune cell activity.	ECS plays a role in stress-induced immunosuppression; potential target for stress-related immune dysfunction.
[159]	ECS regulation of appetite in obesity	Obese mouse model, administration of CB1 receptor antagonist, food intake and weight monitoring.	CB1 antagonist reduced food intake and body weight in obese mice.	CB1 receptor antagonists may be effective in controlling appetite and weight in obesity treatment.
[160]	ECS and cancer cell proliferation	In vitro study on cancer cell lines, treatment with CB1/CB2 agonists, cell proliferation assays.	CB1 activation promoted, while CB2 activation inhibited, cancer cell proliferation.	Differential roles of CB1 and CB2 in cancer cell growth suggest receptor-specific therapeutic strategies.
[161]	ECS involvement in neuropathic pain	Rat model of nerve injury, administration of FAAH inhibitor, pain sensitivity assessment.	FAAH inhibition increased endocannabinoid levels and reduced neuropathic pain behaviors.	FAAH inhibitors may be potential treatments for neuropathic pain.

## Data Availability

No new data were created or analyzed in this study. Data sharing is not applicable to this article.

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
