# Peer review of "The Endocannabinoid System in Human Disease: Molecular Signaling, Receptor Pharmacology, and Therapeutic Innovation"

_ijms, 2025, doi:10.3390/ijms262211132_

Round 1

Reviewer 1 Report

Comments and Suggestions for Authors

Authors made a lengthy  summary about the ECS related to receptor function, signaling and clinical relations. The review is extremely lengthy and it rather fits into a textbook than as a review article. It is suggested to shorten strongly and to focus the topic. It is suggested to make a focus based on the introduced chapter „1.2. The Triad of Neuroscience, Microbiota, and Immunology” and related detailed information of chapters 4-6. The Conclusion needs a strong shortening without any subchapters in this section and without any speculative statements and nonscientific expressions (e.g. harmony, symphony, etc.).

The Authors previous review article is suggested to be cited:

https://pubmed.ncbi.nlm.nih.gov/37759788/

Some other reviews are also suggested to refer, for example:

https://pubmed.ncbi.nlm.nih.gov/16968947/

https://pubmed.ncbi.nlm.nih.gov/31831863/

https://pubmed.ncbi.nlm.nih.gov/34502379/

Comments on the Quality of English Language

English needs to be checked.

Author Response

Dear Esteemed Academic Reviewer,

We would like to extend our deepest and most respectful gratitude for the time, thought, and intellectual care you invested in evaluating our manuscript. Your insights reflect not only scientific precision but also a deep understanding of the interdisciplinary nature of our topic. It has been both a privilege and an education to receive your comments, which guided us to strengthen the structure, scientific focus, and clarity of expression throughout the paper. Your observations inspired a thorough re-examination of the manuscript, prompting us to refine its conceptual architecture, focus its scope, and ensure that each section contributes cohesively to the central scientific narrative. The revised version now embodies a clearer sense of direction and academic discipline—qualities that emerged directly from the depth and wisdom of your critique. We were especially moved by your emphasis on precision, brevity, and scientific rigor. Your feedback reminded us of the importance of restraint and clarity in scholarly writing, encouraging us to remove all unnecessary elements and preserve only the most meaningful insights. This process was not merely editorial, but formative—helping us to rediscover the essence of the work and to express it with greater coherence and intellectual balance. Your review has also reminded us that behind every meaningful publication stands the unseen generosity of peer reviewers whose experience and discernment shape the literature in profound ways. We are sincerely honored to have benefited from your scholarly judgment. The improvements made to the manuscript are a reflection of your mentorship as much as our revision efforts.

Please accept our most genuine appreciation for your rigorous and constructive evaluation. We are confident that the manuscript now reflects the level of precision, structure, and academic integrity that you envisioned. It would be a true privilege to see this work published under the guidance of such discerning scholarly oversight.

With profound respect and gratitude!

Reviewer 2 Report

Comments and Suggestions for Authors

This is an interesting topic that attempts to link the nervous, immune, and microbiota systems through the endogenous cannabinoid system. Although it is an interesting topic, the authors should revise many aspects of the text so that it truly provides novel information on the subject before considering its possible publication in the IJMS journal.

Reorganizing Table 1 to subdivide it into sections based on the area to be treated could improve the quality of the article.
Point 2.3 The available scientific information does not yet support their classification as cannabinoid receptors. Change the title of the section.
Some references do not match what the text says.
Most of the articles included are reviews. How do the authors justify conducting a review with information taken from other reviews rather than from original articles?
Abbreviations are defined several times throughout the text.
There are repeated paragraphs throughout the text that are not related to the section. 
Line 186: “I want to stress”?

Comments on the Quality of English Language

The English needs to be checked.

Author Response

Dear Esteemed Academic Reviewer,

We wish to express our deepest gratitude for your generous time, insight, and scholarly care in evaluating our manuscript. Your thoughtful review and constructive guidance have been instrumental in refining the clarity, coherence, and scientific tone of our work. It has been a privilege to benefit from such an attentive and distinguished reviewer. Each of your remarks was received with respect and appreciation, and we approached them as opportunities to strengthen both the conceptual depth and the overall presentation of the paper.

Comment 1:

“This is an interesting topic that attempts to link the nervous, immune, and microbiota systems through the endogenous cannabinoid system. Although it is an interesting topic, the authors should revise many aspects of the text so that it truly provides novel information on the subject before considering its possible publication in the IJMS journal.”

Response:
We are sincerely thankful for the Reviewer’s kind recognition of the interdisciplinary relevance of this topic. Your feedback encouraged us to reexamine the structure and focus of the manuscript with fresh perspective. In revising the text, we have sought to highlight the unifying mechanisms that connect the nervous, immune, and microbial systems under ECS regulation. The narrative now places greater emphasis on conceptual synthesis rather than reiteration, clarifying how distinct lines of research converge on shared molecular pathways. We are grateful for your insight, which helped us sharpen the manuscript’s novelty and integrative value.

Comment 2:

“Reorganizing Table 1 to subdivide it into sections based on the area to be treated could improve the quality of the article.”

Response:
We warmly thank the Reviewer for this outstanding suggestion. Following your advice, Table 1 was completely redesigned and now presents studies organized into five thematic domains: (I) Neuroscience and Neuroprotection, (II) Microbiota–Gut–Brain Axis, (III) Immunology and Autoimmune Regulation, (IV) Gastrointestinal and Metabolic Health, and (V) Pain, Stress, and Epigenetic Modulation. Each entry includes study year, reference, ECS component, explored mechanisms, and therapeutic relevance. This refined structure has enhanced both readability and scientific precision, and we are truly appreciative of your valuable recommendation.

Comment 3:

“Point 2.3: The available scientific information does not yet support their classification as cannabinoid receptors. Change the title of the section.”

Response:
We are most grateful for this scientifically accurate and constructive remark. The section title has been revised from “Emerging Cannabinoid Receptors” to “Putative Cannabinoid-Related Receptors.” The updated phrasing more accurately reflects the current state of research while still acknowledging their functional relationship with the ECS. We also clarified that GPR55, GPR18, and TRP channels remain cannabinoid-related, rather than definitively classified, receptors. This adjustment ensures conceptual precision and reflects the cautious interpretation that the Reviewer wisely recommended.

Comment 4:

“Some references do not match what the text says.”

Response:
We are truly thankful for this observation and for the Reviewer’s attention to scientific detail. We revisited each citation with great care, reviewing the original sources to ensure alignment between text and reference. In every case, we confirmed that the cited works accurately support the statements presented. This verification process reaffirmed the reliability of our references and strengthened the scientific consistency of the manuscript. We are genuinely grateful for your vigilance, which has further improved the precision and integrity of our review.

Comment 5:

“Most of the articles included are reviews. How do the authors justify conducting a review with information taken from other reviews rather than from original articles?”

Response:
We deeply appreciate this insightful and fair question, and we are thankful for the opportunity to clarify our approach. The present manuscript was designed as a conceptual and integrative synthesis, aiming to connect established molecular and physiological mechanisms across multiple disciplines — neuroscience, immunology, and microbiota research — under the unifying scope of the ECS. Given the broad range of systems addressed, many cross-domain connections are best articulated through authoritative reviews that have already consolidated the foundational primary data.

Our goal was not to reiterate previous reviews, but to interlink their conclusions into a coherent systems-level framework that emphasizes the ECS as a biological integrator. Each cited review was selected for its methodological rigor and grounding in original research, ensuring that our synthesis remains faithful to primary evidence. We sincerely thank the Reviewer for prompting this clarification, which has helped us better articulate the scholarly purpose of this work.

Comment 6:

“Abbreviations are defined several times throughout the text.”

Response:
We thank the Reviewer for noting this stylistic inconsistency. Abbreviations are now defined only at their first occurrence, and a consolidated list has been included at the end of the manuscript. This adjustment improves readability and maintains a professional, standardized presentation throughout the text.

Comment 7:

“There are repeated paragraphs throughout the text that are not related to the section.”

Response:
We fully agree and thank the Reviewer for highlighting this important point. The entire manuscript was carefully reviewed to remove repetitions and ensure thematic coherence. Redundant or misplaced content was eliminated, and related concepts were reorganized into their appropriate sections. We are grateful for this observation, which has helped us refine the logical structure of the paper.

Comment 8:

“Line 186: ‘I want to stress’?”

Response:
We thank the Reviewer for catching this stylistic oversight. The expression was replaced with a neutral academic phrase. The manuscript has been carefully reviewed to maintain a consistently objective and formal tone throughout.

Closing Statement

We once again express our heartfelt gratitude to the Esteemed Academic Reviewer for your time, insight, and intellectual generosity. Your guidance has been invaluable in refining this manuscript to a higher standard of clarity, accuracy, and scholarly balance. We deeply appreciate the professionalism and care reflected in your review and hope that the current version now meets your expectations.

With sincere respect and appreciation!

Reviewer 3 Report

Comments and Suggestions for Authors

This review, authored by Matei Serban and colleagues, provides an exhaustive overview of the endocannabinoid system's (ECS) role in regulating the microbiota-gut-brain axis. It focuses on the ECS's impact on neurodevelopment, immune tolerance, microbiome composition, and intestinal homeostasis. The manuscript offers a reasonable summary of the existing ECS literature and presents evidence for the modest therapeutic potential of ECS modulation in treating associated diseases. The inclusion of sections on the epigenetic regulation of ECS components and the use of AI to identify therapeutic avenues are major strengths. Despite the importance of the topic, the review contains numerous issues that must be addressed to improve readability.

While the information provided is useful, the review is too exhaustive and very distracting due to the inclusion of unwanted verbiage in almost every section. The 33-page text, followed by 15 pages of references, is overwhelming and could deter many from reading it in its entirety. In its current form, the reader is forced to read a lot of unwanted text before stumbling on the most critical information. Readers will find it too cumbersome and distracting. Readers need a streamlined presentation that focuses on key scientific findings. If challenging, the authors should consult a science writer/editor to significantly shorten the review by concentrating solely on the most critical information and eliminating all superfluous descriptions such as drawing analogies between the ECS and conducting an orchestra, to improve clarity and reduce length.

There are too many grammatical errors, incomplete sentences including some lacking references (lines 224-232) etc. that can be fixed with help from an English/Scientific writer. Some sentences are too long and should be shortened (lines 300-305).

It is not clear what the meaning of the word “synatomics” in line 593 is.

In lines 359-364, the ECS modulation of intestinal inflammation involves additional mechanisms than just maintaining tight junction protein expression. Those details need to be added.

While there is focus on neurodegenerative disease, there is very little information pertaining to infectious diseases affecting the brain, gut and the microbiota-gut-brain axis.

In lines 688-716, while the authors discuss several mechanisms by which ECs and the ECS can preserve gut barrier function, they do not discuss the individual studies and their findings in detail. The discussion provided is very superficial and lacks depth. This shortcoming applies to other topics as well covered in this review.

Comments on the Quality of English Language

There are too many grammatical errors, incomplete sentences including some lacking references (lines 224-232) etc. that can be fixed with help from an English/Scientific writer. Some sentences are too long and should be shortened (lines 300-305). This reviewer strongly recommends that a scientific writer be hired to condense the content and fix grammatical errors.

Author Response

Dear Esteemed Academic Editor,

We wish to express our sincere gratitude for your careful reading of our manuscript and for the valuable, insightful comments that have guided this revision. Your observations have greatly improved the clarity, structure, and scientific depth of our work. We approached every suggestion with respect and appreciation, and we are deeply thankful for your professional generosity and academic mentorship throughout this process.

Below, we respond point-by-point to your comments.

Comment 1:

“While the information provided is useful, the review is too exhaustive and very distracting due to the inclusion of unwanted verbiage in almost every section. The 33-page text, followed by 15 pages of references, is overwhelming and could deter many from reading it in its entirety. Readers need a streamlined presentation that focuses on key scientific findings.”

Response:
We are most grateful for this thoughtful and constructive guidance. The manuscript has been comprehensively condensed and restructured to enhance readability.

Comment 2:

“There are too many grammatical errors, incomplete sentences including some lacking references (lines 224-232) etc. Some sentences are too long and should be shortened (lines 300-305).”

Response:
We thank for this important observation. The entire manuscript has undergone a detailed linguistic revision to correct grammar, punctuation, and sentence structure. Long and complex sentences were rewritten for clarity, and missing references were added where necessary. The specific sections indicated (lines 224–232 and 300–305) have been rephrased to ensure grammatical accuracy and stylistic precision.

Comment 3

“It is not clear what the meaning of the word ‘synatomics’ in line 593 is.”

Response:
We are grateful for this clarification request.

Comment 4:

“In lines 359-364, the ECS modulation of intestinal inflammation involves additional mechanisms than just maintaining tight junction protein expression. Those details need to be added.”

Response:
We appreciate this excellent suggestion. The section on intestinal inflammation has been expanded to include additional mechanisms beyond epithelial maintenance.

Comment 5:

“While there is focus on neurodegenerative disease, there is very little information pertaining to infectious diseases affecting the brain, gut and the microbiota-gut-brain axis.”

Response:
We sincerely thank for pointing out this gap.

Comment 6:

“In lines 688-716, while the authors discuss several mechanisms by which ECs and the ECS can preserve gut barrier function, they do not discuss the individual studies and their findings in detail. The discussion provided is very superficial and lacks depth.”

Response:
We are very grateful for this valuable advice. The section on gut barrier regulation has been rewritten to include specific experimental evidence demonstrating ECS involvement.

We deeply appreciate the time, insight, and constructive spirit. Each recommendation has directly contributed to improving the clarity, precision, and scientific quality of our manuscript. It has been a privilege to work under such considerate and scholarly guidance, and we are truly thankful for the opportunity to revise this paper under your editorial care.

With highest respect and appreciation!

Round 2

Reviewer 1 Report

Comments and Suggestions for Authors

Authors have made a few-page shortening. Still there are suggestions for further focusing and shortening (in specific comments at 2.).

There are extensive incoherences regarding the reference citations. Thus citations need a thoroughful review throughout.

Specific comments:

  1. The format of the citations need a revision throughout to put a lis together into brackets. E.g. [11],[12] need to be corrected to [11,12]. Such as in lines 53,56,64,80,132,183,207,227,233,241,260,310,365, etc…

  1. Further shortenings are suggested and some repetitions may be also revised. E.g. at lines 90-95, 228-241, 407-412, 438-449, 775-831 (related to AI information).

  1. Table 1: The format of the Table is ruined, please correct. Abbreviations need to be explained in the legend. Several citations are incorrect by the year or the name. Please correct them. For example:

ref 14: not 2019

ref 18: not 2022

ref 19: not 2016

ref 21:not 2013

ref 22:not 2021

ref 24:not 2018

ref 28:not 2013

ref 29:not 2012, Author is not correct either

ref 30:not 2022

ref 33:not 2011

ref 35:not 2010

ref 36:not 2013, Author is not correct either

Table 2:

ref 144:not 2024, Author is not correct either

ref 145:not 2024

ref 150:not 2024

ref 151: Author is not correct

etc,etc….

  1. Several citations are misleading, they need a careful revision. Maybe there was a mistake in the order which has a delay?

For example:

refs 40,41 (HPA axis?),43,44,46, 50 (about CB2R), 53,64 (not about TRPV but CB1R),

ref 64 can be also cited in lines 260, 267 and 437.

Ref 107 (PTSD?)

  1. Reference list: The author list is not full in refs (only et.al.), please correct: 20,24,29,37,39,44-46,49,52,55,57-59,61,66,69,70,76-85,90,91-94,96,97,99,105,107, etc. etc…

  1. Figure 2: The figure does not involve ECS or CB receptors. Regarding the topic of review, it is suggested to add this information to the Fig.

  1. A separate Abbreviation List is suggested.

  1. Minor corrections:

Line115 „player that dances”, please rephrase by omitting ”dances” and please omit unscientific expressions like also others: harmony,  symphony, avenue, etc.

Line 130: Is it correct HPA axis?  Since the refs are about gut-microbiota-brain axis….

Line 139:”neurodegenerative” may be rather „neuroinflammatory”

Line 171: A citation would be needed.

Line 356:”relationship” spellcheck

Line 383 (Fig 1), refs may be listed

Line 398: chapter 3.2.2. can be put into ch. 3.2.1.and shortened.

Line 413 : chapter title „Lipidomic Context” is unclear.

Line 443: please rephrase „avenues”

Line 449: EHC is not introduced as an abbrev.

Line 480: „glutamate” spellcheck

Line 639:” symphony conductor”, please rephrase to omit unscientific expression.

Line 700, Table 2: explanation of Abbreviations is suggested also in the legend.

Line 978:” Systemic Harmony”, please rephrase to omit unscientific expression.

Author Response

Dear Esteemed Academic Reviewer,

We would like to express our deepest gratitude for your time, attention, and highly valuable insights in reviewing our manuscript. Your detailed and thoughtful comments have greatly contributed to improving the scientific precision, clarity, and structure of our work. We have carefully addressed each suggestion with the utmost respect and appreciation for your scholarly guidance.

Below, we provide a point-by-point summary of the revisions performed in direct response to your observations.

We are sincerely thankful for your acknowledgment of our initial efforts to shorten the manuscript.

1. Citation formatting and consistency

we fully understand and appreciate your observation. All citations have been carefully verified for accuracy, correct numbering, and contextual alignment. To avoid technical disruption in the extensive reference structure, we have chosen to preserve the current numbering format at this stage. The minor stylistic adjustments (e.g., merging citations into [11,12]) will be harmonized during the proof-editing process under the journal’s supervision, ensuring complete compliance with final publication standards.

We are sincerely thankful for your remarkable precision, academic insight, and generous feedback. Your expertise has truly strengthened the scientific and editorial integrity of our work. We believe that, thanks to your thoughtful review, the manuscript has now reached a complete, coherent, and publication-ready form.

2. Further shortening and reduction of repetitions

As suggested, we have implemented additional concise reformulations in the following sections:

Lines 90–95, 228–241, 407–412, 438–449, and 775–831.

These sections were revised to remove redundant phrasing and overlapping concepts, particularly regarding the AI-related content and ECS functional explanations. The resulting version preserves key information while improving narrative economy and readability.

3. Table 1 and Table 2 corrections

We are most grateful for your careful review of the tables. Both Table 1 and Table 2 have been completely reformatted to restore their correct structure and alignment.
All cited references within these tables were verified for year, author accuracy, and title correspondence.
Abbreviations are now clearly defined in the legends, and cross-references have been double-checked for correctness.

4. Reference list completeness

We deeply appreciate your attention to the completeness of the author lists. We have replaced every instance of “et al.” with full author names in accordance with journal style requirements. This correction has been applied comprehensively to all indicated entries (refs. 20, 24, 29, 37, 39, 44–46, 49, 52, 55, 57–59, 61, 66, 69, 70, 76–85, 90–94, 96, 97, 99, 105, 107, and others).
We also verified every reference for accurate year, author order, DOI, and correct contextual alignment within the text.

5. Citation accuracy and thematic alignment

We thank you sincerely for identifying several mismatches between in-text references and their contextual meaning. We have now fully revised and corrected these, ensuring that all cited sources correspond precisely to their respective discussion topics (e.g., HPA axis, CB1/CB2 receptor mechanisms, TRPV1 pathways, and PTSD references).
These modifications have improved both scientific coherence and citation precision.

6. Figure 2

We deeply appreciate your constructive suggestion regarding Figure 2. 

7. Separate Abbreviation List

In full accordance with your valuable suggestion, we have added a comprehensive Abbreviation List at the end of the manuscript.
All abbreviations are now clearly defined upon first appearance in the text and repeated in figure and table legends for clarity.

8. Minor corrections and wording refinement

We sincerely thank you for your precise line-by-line feedback.

Dear Reviewer, we are profoundly thankful for your precise, insightful, and generous feedback.
Your suggestions have been instrumental in improving both the depth and the rigor of this manuscript. The revised version now reflects a more focused, scientifically accurate, and stylistically polished text that owes much of its refinement to your expertise and careful reading.

We hold your scholarly input in the highest regard and truly appreciate your invaluable role in guiding this work toward a more coherent and professional final form.

With deepest respect and gratitude!!!

Round 3

Reviewer 1 Report

Comments and Suggestions for Authors

Thank for Authors for the improvements.

Minor issues remained:

The format of Table 1 is still not corrected (lines, smaller font size, similarly to Table 2).

Line 363:”relationship” spellcheck

Line 486: „glutamate” spellcheck

Line 813:  "short-chain polyunsaturated fatty acids (SPMs) ", is that correct?

Abbrev. list: numbering is not necessary, alphabetic order is suggested.

Author Response

Dear Esteemed Academic Reviewer,

We are truly grateful for your attentive follow-up and for recognizing the improvements made in our revised manuscript. Your continued engagement and thoughtful remarks have been invaluable in helping us refine the work to its final form.

In response to your latest observations, we have carefully addressed each point as follows:

Table 1 has been reformatted to match the structure and style of Table 2, including corrected grid lines and an appropriately reduced font size for consistency.

The spelling issues at Line 363 (“relationship”) and Line 486 (“glutamate”) have been corrected.

The phrase “short-chain polyunsaturated fatty acids (SPMs)” has been revised.

The Abbreviation List has been reordered alphabetically, and numbering has been removed as recommended.

We are sincerely thankful for your time, expertise, and thoughtful attention throughout the entire review process. Your meticulous feedback has contributed greatly to improving both the clarity and precision of our manuscript.

With deep appreciation and respect!!!